



# A novel measurement system for unattended, in-situ characterisation of carbonaceous aerosols

Alejandro Keller, Patrick Specht, Peter Steigmeier, and Ernest Weingartner

University of applied sciences and arts northwestern Switzerland, Klosterzelgstrasse 2, 5210 Windisch, Switzerland

**Correspondence:** alejandro.keller@fhnw.ch

**Abstract.** Carbonaceous aerosol is a relevant constituent of the atmosphere in terms of climate and health impacts. Nevertheless, measuring this component poses many challenges. There is currently no simple and sensitive commercial technique that can reliably capture its totality in an unattended manner, with minimal user intervention, for extended periods of time. To address this issue we have developed the fast thermal carbon totalizator (FATCAT). Our system captures an aerosol sample
on a rigid metallic filter and subsequently analyses it by rapidly heating the filter directly, through induction, to a temperature around $800°C$. The carbon in the filter is oxidized and quantified as $CO_2$ in order to establish the total carbon (TC) content of the sample. The metallic filter is robust, which solve filter displacement or leakage problems, and does not require a frequent replacement like other measurement techniques. The limit of detection of our system using the $3\sigma$ criterion is $TC = 0.19\,\mu g\text{-C}$ (micrograms of carbon). This translates to an average ambient concentration of $TC = 0.32\,\mu g\text{-C}/m^3$ and $TC = 0.16\,\mu g\text{-C}/m^3$
for sampling interval of one hour or two hours respectively using a sampling flowrate of 10 lpm. We present a series of measurements using a controlled, well defined, propane flame aerosol as well as wood burning emissions using two different logwood stoves. Furthermore, we complement these measurements by coating the particles with secondary organic matter by means of an oxidation flow reactor. Our device shows a good correlation (correlation coefficient, $R^2 > 0.99$) with well-established techniques, like mass measurements by means of a tapered element oscillating microbalance and TC measurements by means
of thermal optical transmittance analysis. Furthermore, the homogeneous fast-heating of the filter produces fast thermograms. This is a new feature that, to our knowledge, is exclusive of our system. The fast thermograms contain information regarding the volatility and refractoriness of the sample without imposing an artificial fraction separation like other measurement methods. Different aerosol components, like wood burning emissions, soot from the propane flame and secondary organic matter, create diverse identifiable patterns.

## 1 Introduction

Carbonaceous aerosols are a minor constituent of the atmosphere by mass, but a critical component in terms of impacts on the climate and especially climate changes. Several of its properties are considered core aerosol properties by Global Atmosphere



Watch (GAW) and essential climate variables (ECV) by the Global Climate Observing System (GCOS) (Laj et al., 2020).
At the same time, estimates suggest that particulate matter pollution, which largely composed of carbonaceous material, is responsible for one of every 13 premature deaths (Fuller et al., 2022) and the World Health Organization has classified diesel exhaust (a major source of carbonaceous aerosols) as carcinogenic to humans. The size of the particles is very relevant as it directly influences physical and chemical properties. Particles with diameters smaller than 1 µm are of special concern because they live longer in the atmosphere, penetrate deeper into the human respiratory system, and are composed of materials that are
climate and health relevant. In particular, carbonaceous material from biogenic and anthropogenic sources is usually the largest aerosol fraction in this size range. It accounts for 50 to 70% of the particles with diameters smaller than 1 µm in polluted and pristine areas (Szopa et al., 2021). Comprehensive long-term measurements of aerosol composition and physical properties are of paramount importance for assessing aerosol effects on climate and health and for devising effective mitigation strategies. However, there is still no commercial instrument that can measure the totality of carbonaceous aerosol with sufficient accuracy
and temporal resolution on a global level over extended periods of time in an unattended manner with minimal user intervention. As a consequence, knowledge of the atmospheric abundance of carbonaceous aerosol relies on approximate models that provide estimates with low confidence and global trends cannot be characterized due to limited observations (e.g. Szopa et al., 2021).

The term carbonaceous aerosols comprise very diverse substances with a continuum of properties (thermal, optical, etc.) and various degrees of toxicity (Pöschl, 2005). This complexity has created a desire to split carbonaceous aerosols into fractions in
order assess its true impact as well as to understand atmospheric cycles, including the formation of secondary organic aerosol (SOA). On a first approximation, carbonaceous aerosols are classified in terms of composition as organic and inorganic. Per definition, only organic aerosol (OA) contains carbon–hydrogen bonds. Nevertheless, pure inorganic carbonaceous aerosols are rare. Even soot and commercial carbon black materials used as reference aerosols for calibration purposes contain hydrogen and other elements (Clague et al., 1999). Hence, it is way more common to express OA through a carbon mass fraction called
organic carbon (OC), defined pragmatically as the complementary carbon mass fraction to equivalent black carbon (eBC) or elemental carbon (EC), which are measured by light absorption or thermal refractivity, respectively. These definitions are not interchangeable, as EC cannot be unequivocally inferred from eBC. Another possibility is to measure the complementary fraction of OC as refractive black carbon (rBC) through laser-induced incandescence (Stephens et al., 2003). This quantity is once more not interchangeable with neither eBC nor EC. Although there are commercial instruments available for measuring
rBC, for the purpose of simplicity, we will limit the current discussion to eBC, EC and OC.

Thermal refractivity methods are performed off-line or semi-online using samples captured on a filter (Cavalli et al., 2010, and references therin). The analysis process consists of two main steps. The first step, performed under an inert atmosphere, targets the OC fraction whereas the second step, performed using an oxidizing gas mixture, targets the EC fraction. The process is defined by standard temperature protocols that further divide both fractions into "ideal" subfractions selected according to the
properties of ambient samples from specific regions. Different protocols vary in terms of the number of temperature set-points that define the subfractions as well as on the duration of the measurement time at each set-point. For instance, there are marked differences in two of the thermal-optical protocols most widely used by the atmospheric science community. The EUSAAR2 protocol considers 4 subfractions for OC and 4 for EC and has a total analysis time of 17 minutes, whereas the IMPROVE





protocol considers 4 subfractions for OC and 3 for EC and has a variable analysis duration between 17.5 and 67 minutes. These

methods are prone to artifacts. Even the determination of the split point between EC and OC fractions is difficult to determines as it depends upon several factors (Panteliadis et al., 2015). A main source of uncertainty is the production of pyrolytic carbon (PC) from OC during the inert-gas analysis-step. This artefact can be compensated to some degree by using a thermal-optical correction, which involves monitoring the filter sample using light transmission (i.e., using thermal optical transmission, TOT) or light reflection (i.e., using thermal optical reflectance, TOR). Without correction, PC is wrongly assigned to EC.

Efforts to reduce discrepancies in the OC–EC fraction separation of thermal-optical measurements using an enhanced temperature calibration during a round robin comparison resulted in a moderate improvement, with a repeatability and reproducibility of the order of 20% for the EC fraction when using the same thermal protocol (i.e., EUSAAR2 or NIOSH870) and the same PC correction strategy (i.e., TOR or TOT; Panteliadis et al., 2015). Nevertheless, the variation in the estimation of the EC fraction was as high as 113% when comparing different protocols and/or different correction strategies. Problems like

dependency of measurement day, variations in flow-rate within the accepted operation range, variations in the calibration gas (i.e. when changing the gas bottle) or in transit time through the instrument, leakages, and different rates of pyrolyzed OC production were reported as sources of unresolved systematic errors. These results question the significance of an OC–EC split using currently available thermal-optical analysis systems. Interpretation of the OC subfractions is also not straightforward, as they do not provide a clean separation of OC in terms of molecular components or volatility (Diab et al., 2015). Light absorp-

tion methods for measuring eBC are also prone to systematic errors (e.g. Weingartner et al., 2003; Collaud Coen et al., 2010). Furthermore, a fraction of OC called brown carbon (BrC) also absorbs solar radiation and contributes together with eBC to a positive radiative forcing of the atmosphere. Although thermal refractivity methods and light absorption methods are established monitoring techniques used extensively by the scientific community, their measurement artifacts and the impossibility to establish a strict separation point between fractions limit their usability as long-time monitoring techniques.

Traditionally, long-term measurements of chemical composition have been made through the periodic (e.g., daily or weekly) collection of filter samples, followed by offline chemical analysis (e.g. Chow, 1995; Müller et al., 2004). These methods have provided valuable long-term data that have been crucial for identifying multi-year trends in ambient aerosol composition. However, their low time-resolution reduces the efficacy of source apportionment techniques in comparison to online instrumentation, and they may suffer from artifacts relating to the collection and/or storage of reactive or semivolatile species such

as organics and nitrate (e.g. Zhang and McMurry, 1992; Cheng and Tsai, 1997; Resch et al., 2023).

This manuscript describes the fast thermal carbon totalizator (FATCAT), a new measurement system for unattended long-term measurements of aerosol-bound total carbon. TC seems to be the appropriate metric for a system like this, as it has proven to be a more reliable and reproducible than the split into carbonaceous fractions (Schmid et al., 2001; Haller et al., 2019). FATCAT is simpler, more stable and robust when compared to other techniques. It capture particles on a rigid, long-lived

metallic filter that does not cause leaks or displacement errors which affect field-instruments that utilize soft quartz filters. sample analysis happens in-situ using a short cycle, less than one minute, that generates fast thermograms. This feature needs to be studied further, but our measurements show that these thermograms contain information about the aerosol composition, which could be used for source apportionment studies.



Another instrument for measuring TC, the Total Carbon Analyzer (TCA08, Magee Scientific), has been commercially avail-
able for a couple of years. There are a few differences between FATCAT and TCA08. The TCA08 has a double sampling
head for uninterrupted sampling, does not require a special analysis gas, and collects samples using quartz filter that needs to
be replaced regularly. The current FATCAT prototype has a single sampling head, which needs to cool down before the next
measurement cycle, requires $CO_2$ free and VOC free synthetic air for the analysis, collects samples on a robust long lived
metallic filter, and has a oxidation catalytic converter before the $CO_2$ measurement in order to ensure that all OC will be taken
into account. Other differences include heating strategy (indirect in the case of the TCA08 and direct through induction in
FATCAT) and calibration procedure (model substances for the TCA08 vs. $CO_2$ and mass flow controller calibration in FAT-
CAT). Finally, there are currently no reports on the possibility of generating thermograms with the TCA08. The manufacturer
of the TCA08 suggests using their device in combination with a eBC monitoring device in order to infer sub-fractions based
on two new concepts, the equivalent organic carbon (eOC) and equivalent elemental carbon (eEC), that rely upon regional and
seasonal calibration (Rigler et al., 2020).

## 2 Experimental Setup

### 2.1 The fast thermal carbon totalizator (FATCAT)

FATCAT is prototype instrument designed and constructed by the FHNW for in-situ measurement of carbonaceous aerosol.
Figure 1 shows the flow diagram of the instrument. The system has three inlets: zero air, sample, and bypass. The sample and
bypass inlets are connected to internal three-way valves actuated in such a way that only one of them is open at any given
time, allowing to choose between the sampling and analysis operation mode. The bypass inlet is used for applications where
the instruments needs to draw a constant amount of air from a sampling head even during the sample analysis. This makes the
instrument compatible with measurement stations that have size selection sampling heads (like, PM1, PM2.5 and so on) that
require defined flowrates.

FATCAT is build as a stand-alone measurement system, which does not require external laboratory equipment (other than
the optional external vacuum pump and denuder). The status data and all relevant parameters can be read through a USB serial
interface, which also serves as the link for sending commands. Parameters can also be adjusted and monitored directly at the
device through an LCD display accessible throughout a user menu using the interface buttons. The timing of sampling and
analysis cycles and data logging are performed by a Raspberry Pi 4B microcomputer (Raspberry pi foundation, UK) using
software programmed in python. The software provides a graphical user interface, but the device can also run "headless" using
programmed scripts.

During sampling, the instrument opens the sample inlet and closes the sample bypass. In this mode FATCAT gathers an
aerosol sample on a sintered hastelloy-X filter (SIKA-HX3; GKN sinter metal filters, Germany). When using only the internal
pump, the sampling flowrate can be regulated up to a maximum of 2 lpm. An external pump can be used for applications that
require higher flows rates. In this configuration, the sampling flowrate is constrained by atmospheric pressure, as the sintered
filter acts as an ensemble of critical orifices. Notably, the maximum achievable sampling flowrate is typically around 10 lpm at





the Swiss plateau (approximate elevation of about 400 meters above sea level [m.a.s.]), and 7 lpm at the Sphinx observatory of the Jungfraujoch (situated at 3500 m.a.s.l.). The sample flow throughout the pumps is controlled by two mass flow controllers (MFC; Vögtlin Instruments, Switzerland).

Conversely, the analysis mode seals the sample inlet and opens the sample bypass and zero air inlets. As a standard, we use synthetic air with low carbon dioxide and hydrocarbon content ($CO_2 \leq 0.5$ ppm, hydrocarbons $\leq 0.1$ ppm; Aphagaz 1 synthetic air; Carbagas AG, Switzerland) at a flowrate of 1 lpm. This mode can also be used instead of the sampling mode in order to gather a blank probe to determine the zero offset and its variability. During sample analysis, the induction furnace is turned on and the filter is heated in less than one minute to a temperature of the order of $800°C$. Under the zero air atmosphere, this temperature is enough to effectively oxidize and desorb carbonaceous material collected during the sampling phase. A platinum catalyst (OST.1700.200.A9; Hug Engineering, Switzerland) positioned downstream of the induction furnace and heated to $200°C$ ensures complete oxidation of organic substances and avoids measurement artifacts arising from the incomplete combustion of the sample. This type of catalyst is used for after-treatment of diesel-vehicles emissions, which makes it a very robust and long-living component. A nondispersive infrared (NDIR) carbon dioxide sensor (LI-850; LICOR Germany) is used for $CO_2$ quantification. Sampling can be restarted once the filter cools down to a predefined temperature. A cooling period of approximately 20 minutes is typically required to reach a target temperature of $30°C$. The heating of the filter has been optimized through finite element calculations, ensuring uniform and localized heating of the sample. Several pt-1000 sensors monitor the temperature of the sampling filter, the induction coil, and the catalyst.

Ideally, the total carbon mass in the sample, $m_{\mathbf{TC}}$, can be derived from the $CO_2$ mass concentration, $c_{\mathbf{CO_2}}$, and the mass flowrate, $f$, throughout the instrument using:

$$m_{\mathbf{TC}} = \int_{t_1}^{t_2} \frac{\mathbf{d}m_{\mathbf{TC}}}{\mathbf{d}t} \mathbf{d}t$$

$$m_{\mathbf{TC}} = \frac{M_{\mathbf{C}}}{M_{\mathbf{CO_2}}} \int_{t_1}^{t_2} f(t) \left( c_{\mathbf{CO_2}}(t) - c_{\mathbf{CO_2,baseline}} \right) \mathbf{d}t, \tag{1}$$

where $c_{\mathbf{CO_2,baseline}}$ is the $CO_2$ mass concentration from the zero air, $t_1 \leq t \leq t_2$ is duration of the analysis, and $M_{\mathbf{C}}$ and $M_{\mathbf{CO_2}}$ are the molar masses of C and $CO_2$ respectively. Nevertheless, the heating of the filter causes the filter pores to reduce in size and, as a result, the pressure downstream of the filter will drop. This fast change in pressure is not compensated fast enough by the $CO_2$ sensor, which results in an offset of the $m_{\mathbf{TC}}$ or a non-zero $m_{\mathbf{TC}}$ for a blank sample. The shift in $CO_2$ concentration for a blank sample is, however, reproducible. This allows us to calculate a corrected total carbon mass of the sample, $m_{\mathbf{TC}}^*$, by subtracting an average blank $CO_2$ offset curve from the measured $CO_2$ mass concentration. Equation 1 becomes:



$$m_{\mathbf{TC}}^{*} = \int\limits_{t_1}^{t_2} \frac{\mathbf{d}m_{\mathbf{TC}}^{*}}{\mathbf{d}t}\mathbf{d}t$$

$$m_{\mathbf{TC}}^{*} = \frac{M_{\mathbf{C}}}{M_{\mathbf{CO_2}}} \int\limits_{t_1}^{t_2} f(t)\,(c_{\mathbf{CO_2}}(t) - c_{\mathbf{CO_2,baseline}} - \overline{c}_{\mathbf{CO_2,blank,offset}}(t))\,\mathbf{d}t, \tag{2}$$

where $\overline{c}_{\mathbf{CO_2,blank,offset}}(t)$ is the average evolution of the $CO_2$ mass concentration for a blank sample. This parameter can be expressed as

$$\overline{c}_{\mathbf{CO_2,blank,offset}}(t) = \frac{1}{n}\sum_{i=1}^{n}\left[c_{\mathbf{CO_2,blank},i}(t) - c_{\mathbf{CO_2,baseline,blank},i}\right], \tag{3}$$

for $n$ analysed blanks with $CO_2$ curves $c_{\mathbf{CO_2,blank},i}$ and baselines $c_{\mathbf{CO_2,baseline,blank},i}$. It is important to include the individual baseline levels of $CO_2$ in the calculation to account for variations of the zero gas composition or long-term drifts of the $CO_2$ sensor.

Finally, the average mass concentration of total carbon, $c_{\mathbf{TC}}$, after sampling a volume, $V$, of carrier gas can be calculated from the total carbon mass in the filter as $c_{\mathbf{TC}} = m_{\mathbf{TC}}^{*}/V$.

### 2.2 Baseline

We performed a series of periodic blank measurements to determine the offset of our system. For this purpose, we used FATCAT to sample ambient aerosol from the exterior of our laboratory with a flow-rate of 10 lpm. The campaign started with a new filter that was exposed to ambient aerosols during the day. The ambient sample was analyzed in one hour intervals. Once a day, after the sample analysis at 11pm, the ambient sample was automatically replaced with a blank measurement as described

in the previous section. The purpose of this exercise was to investigate the drift of instrument baseline during its deployment at a measurement site. Drift of the $CO_2$ sensor as well as other factors like contamination of the instrument and obstruction of the filter may influence the results. Blank measurements are also used to calculate the average offset curve described in equation 3. The setup for ambient measurements and examples of measurements at different locations is beyond the scope of this manuscripts and will be discussed in a future publication.

### 2.3 Aerosol generation in the laboratory


We tested FATCAT using carbonaceous aerosol from a combustion aerosol standard generator (CAST; Jing Ltd, Switzerland). Figure 2 shows a schematic representation of the experimental setup. Two main sets of experiments were performed. The first set of experiments consisted of the aerosol sample directly produced by the generator. The CAST generates aerosol particles using a quenched propane diffusion flame. The fraction of OC and EC in the particles varies depending on the settings, in

particular the air to fuel ration during combustion. The second set of experiments consisted of CAST particles with a high



EC-to-TC ratio, which were then coated with different amounts of secondary organic matter (SOM), produced using $\alpha$-pinene ($\geq$ 97% purity, Sigma Aldrich, Switzerland) as a precursor substance, by means of an organic coating unit (OCU Keller et al., 2022). This set of experiments was performed during a separate campaign. The experimental setup is described in detail by (Kalbermatter et al., 2022). The CAST set-points used for both campaigns can be found in table 1. The sample was diluted by

means of a homemade rotating disc diluter (Hueglin et al., 1997) using synthetic air as a carrier gas (Aphagaz 1 synthetic air; Carbagas AG, Switzerland). A three way valve was used to select between the aerosol sample and the particle free synthetic air. This was done to ensure that the measurement devices and the sampling filter were exposed to the same concentration of aerosol particles during the same interval of time.

## 2.4 Characterization of laboratory samples

A tampered oscillating micro balance (TEOM model 1405; Thermo Fischer Scientific Inc., USA) with a flowrate of 1.5 lpm was used to measure the mass concentration of the sample. The temperature of the TEOM sampling head was set to 50°C for samples C1 through C3 and to 30°C for the M1 samples of the coating experiments to minimize the desorption of coating material. The oscillating frequency of the microbalance was logged by a computer every 10 seconds throughout serial connection. The mass increment of the oscillating element was calculated using this frequency and the calibration constant of the TEOM

as described by the user's manual. On a sampling line parallel to the TEOM, an active charcoal denuder (Part. No. M3456; Aerosol d.o.o., Slovenia) was used to remove gas-phase species to avoid positive sampling artefacts. This procedure was not applied to the TEOM as its filter is heated. Aerosols were collected downstream of the denuder by FATCAT at a rate of 1.5 lpm and on quartz fiber filters (filter diameter 47 mm) at a rate of 1 lpm for TOT analysis using the EUSAAR2 protocol. Samples from the coating experiments were analyzed by the Swiss Federal Institute of Metrology (METAS), whereas the rest of the

samples were analyzed by a commercial laboratory (Particle Vision GmbH, Switzerland). The TOT analysis returns the OC and EC concentration per square centimeter of the sample, divided into different subfractions. The total amount of OC, EC and TC capture in the filter can be calculated from this fractions. The data from the TOT analysis was adjusted for flowrate and filter surface for comparison against the TC measurements from FATCAT.

## 2.5 Biomass burning samples

We performed a set of experiments with biomass burning samples in order to challenge FATCAT with high loads of an aerosol with variable, not entirely carbonaceous composition. The tests were performed according to the EN 16510 standard series for residential solid fuel burning appliances at the certified biomass combustion test bench of our partner institute of bioenergy and resource efficiency FHNW. The tests were complemented with our own measurements of TC using FATCAT and filter collection for thermal-optical analysis following the procedure described by Keller and Burtscher (2017). Shortly, a partial

flow was taken from the stack and diluted at a factor 1:4 using zero air at a temperature of 200°C. The purpose of this dilution is to increase the oxygen content and avoid condensation of water once the sample cools down at room temperature. The diluted flue-gas is then cooled down to room temperature and aged by means of an oxidation flow reactor (i.e., the micro smog chamber, MSC; Keller and Burtscher, 2012) in order to promote the formation of secondary organic aerosols. Samples for TOT



analysis are gathered downstream of the MSC. We modified the original setup to include sampling by FATCAT in parallel to the TOT samples. The flow through the MSC was held at a rate of 1 lpm, but the sample was diluted using additional zero air downstream of the MSC. The goal was to create a total of 4 lpm, that could then be sampled in parallel by FATCAT and on a quartz fiber filter. The quartz filters for TOT analysis sampled emissions using a flowrate of 1 lpm, whereas flows of 1, 2 or 3 lpm were used to collect samples in FATCAT. This was done in order to challenge the instrument with high filter loads. Here as well, the results from the TOT analysis were adjusted for flow and filter area to compare them against FATCAT.

Table 2 shows characteristics of the stoves selected for the experiments. The first one is a modern, certified stove for cooking and baking. The second one is an old stove model that has been discontinued by the manufacturer. Three cycles were performed each measurement day. We measured the first cycle of the day (i.e., cold cycle), which was then followed by two immediate warm refueling cycles. Each cycle takes approximately 40 minutes. We only measured the second warm cycle due to the 20 minutes recovery time need for the FATCAT filter to reach $30°C$.

## 3 Results and discussion

### 3.1 Long term behavior of the baseline

Figure 3 shows the result of a periodic, once a day, blank measurement for a total of 109 days, where the FATCAT sampled and analyzed zero air. The purpose of this exercise was to determine the long-time stability of the system. NDIR sensors, like the one used by FATCAT, are very precise in the short term but suffer from offset drifts over extended time periods. Nevertheless, for the purpose of our measurements, we only require the $CO_2$ signal to be stable for less than 120 seconds. The $CO_2$ concentration in the zero-air, measured before the start of the analysis, builds the baseline for the measurement. Still, factors like ambient pressure and temperature, contamination of the sampling lines with organic material, or deterioration of the filter could affect the long time performance.

The curves presented in figure 3a and 3b show that the evolution of temperature, pressure was very reproducible during this prolonged campaign. The temperature evolution affects how the sample will be released by combustion or desorption from the filter whereas the pressure affects the optical measurement of the $CO_2$ concentration. The increase in pressure drop is caused by the reduction of the pore size due to the expansion of the metal of the filter during heating. Ideally, this change in pressure would be compensated by the correction algorithm of the $CO_2$ sensor. A constant $CO_2$ concentration would result in a constant $CO_2$ signal independent of the pressure drop, Nevertheless, figure 3c shows that this is not the case as the differential TC signal diverges from the zero-line for the zero air measurement. This is most likely caused by a too slow pressure compensation of the $CO_2$ sensor, which is design for ambient conditions where fast changes in pressure are not expected. We are considering accessing the raw extinction signals from the $CO_2$ and constructing our own correction algorithm or contacting the manufacturer for an OEM solution for a future optimization of our measurement system.

The offset of the total carbon signal of individual blank measurements (figure 3d) is calculated by integrating the evolution of total carbon, $dm_{TC}$, according to equation 1. As discussed, the integral of a blank sample is non-zero due to the pressure drop in the $CO_2$ sensor. Longer integration times cause larger offset (figures 3d and 3e and table 3) and a more dispersed data.



Thus, an optimal limit of detection can be achieved by choosing the shortest possible integration time that still captures the total carbon information from the sample. In our experience, an integration time of 65 seconds is enough for an ambient sample even at an urban location. The corresponding limit of detection would be $TC = 0.19$ µg-C (micrograms of carbon) sampled in the filter. This translates to an average ambient concentration of $TC = 0.32$ µg-C/m$^3$ and $TC = 0.16$ µg-C/m$^3$ for one or two hours of sampling, respectively, using a flowrate of 10 lpm. Lower limits of detection can be achieved through longer sampling periods or, if possible, by using a higher sample flowrate.

### 3.2 Concentration-response analysis

Figure 4a shows a comparison between the aerosol mass measurement by the TEOM and the total carbon mass measured by FATCAT. Samples C1 and C2 were produced by the CAST with a lean flame composition, with an air-to-fuel mixture of C/O=0.26 and C/O=0.25 respectively, which results in particles with a high elemental carbon fraction of $EC/TC = 91\%$ (see table 4). There is an extremely good correlation ($R^2 = 0.999$) between these two instruments based on very different measuring principles. The TEOM measures mass based on the change of the oscillation frequency of a mass transducer and is independent of the particle composition. By definition, elemental carbon consists exclusively of carbon atoms and, thus, this fraction should be measured equally by the two instruments. The slope of the correlation, $m = 0.94$, is very close to the EC/TC=0.91 calculated using TOT. The difference could be explained by the carbon mass provided by the OC fraction of the samples. The insert figure 4a shows the third sample, C3, which was produced with a rich flame (C/O=0.41) which increases the OC fraction in the aerosol. The correlation between FATCAT and the TEOM is also good. However, the intercept is shifted from zero and the slope of the curve is steeper than for the C1 and C2 samples. The latter is unexpected given that OC is not exclusively composed of carbon. A positive artifact from the TEOM, due to the retention of gas phase organic species, could cause the displacement of the intercept and the steeper slope. The TEOM minimizes positive sampling artifacts by heating the sampling filter to a standard temperature of 50°C, this targets humidity and may even cause negative artifacts from the most volatile fraction of OC. However, this temperature may not be enough to prevent the adsorption of gas-phase OC form the C3 sample which contains mainly species detected during the OC3 step (i.e., desorption at 250°C under an helium atmosphere) of the TOT analysis (figure 4c). A negative artifact from the side of FATCAT cannot be discarded, but is less likely to happen due to the low volatility of the OC3. Additionally, a negative artifact would not explain the displacement of the intercept, as the artifact would be more pronounced at higher filter loads due to the longer sampling time required to achieve them. This would result in a less pronounced slope for this sample.

The induction-based flash-heating furnace of FATCAT allows for a direct, fast and homogeneous heating of the filter. This type of heating produces reproducible $CO_2$ signal-patterns that depend on the sample composition and, thus, can be used to extract information beyond TC quantification. We refer to them as fast-thermograms because they resemble thermograms produced by thermal-optical methods. Nevertheless, there is neither a heating protocol based on predefined temperature steps nor a split between EC and OC using different gases. The use of an oxidizing atmosphere during the heating process prevents the production of pyrolytic carbon, which is a main source of uncertainty in thermal-optical methods. Figure 4b shows the thermograms for the C1 through C3 samples. Samples C1 and C2 where produced with similar air-to-fuel mixing ratios, have a





comparable EC/TC fraction and are also very similar in terms of the OC and EC subfractions from the TOT analysis (figure 4c). Sample C3 was created with a richer flame and has a higher OC fraction (35%). The EC fraction evolves later in the analysis, at higher filter temperatures, than the OC fraction. The two samples with high EC/TC (i.e., C1 and C2) create different patterns even though they were created with similar air-to-fuel mixtures. C2 appears to be more homogeneous. It has a narrow and well

defined distribution. The main difference between the samples is the residence time in the flame before quenching which was shorter in the case of C1 compared to C2. This leads to smaller particles with $GMD = 72$ nm and $GMD = 150$ nm for C1 and C2 respectively. Soot formation models suggest that variations in both particles size and flame carbon-to-oxygen have an effect in the degree of maturity of soot particles (Kelesidis and Pratsinis, 2019b). This has been validated for particles from the CAST generator (Kelesidis et al., 2017, 2021). Mature soot particles have smaller oxidation rates than more nascent ones, as

they contain less hydrogen (see, e.g., Kelesidis and Pratsinis, 2019a). These difference may explain the broadening and shift of the thermograms.

Figure 5 shows the results from the coating experiment. We start with an uncoated seed aerosol (M1) sample with an elemental carbon fraction $EC/TC = 0.84$ and a size distribution with $GMD = 84$ nm. The particles are gradually coated in three steps (coatings 1 through 3) by SOM from the ozonolysis of $\alpha$-pinene. The particle size increases due to the addition

of organic material up to $GMD = 126$ nm, while the elemental carbon fraction is reduced to $EC/TC = 0.1$. Carbon is only a fraction of the coating material, as SOM produced by ozonolysis also contains hydrogen and oxygen. Thus, the mass fraction of carbon to the total mass of the particles is reduced by every further coating step (figure 5a). As an example, by looking at the oxidation state of carbon, we can determine that carbon may amount to only 55% of the total mass of $\alpha$-pinene SOM produced using a similar setup (Leni et al., 2022, supplementary information). Other oxidation states, leading to diverse contributions

of carbon to the total aerosol mass, are also possible (see, e.g., Cain et al., 2021). In our case, most of the organic material corresponds to OC1, the most volatile fraction of the thermal-optical analysis (figure 5c). This step is performed at 200°C under an helium atmosphere. Nevertheless, there is also additional material in all the other organic carbon analysis steps, which speaks for a diversity of organic species. The concentration of seed particles and, thus, the amount of EC was kept constant throughout the different coating steps. The fast thermograms (figure 5b) show that, for this internally mixed particles,

the organic material adds an independent feature at lower temperatures in the thermogram without affecting the shape of the contribution of the uncoated seed. This is not evident as the organic material causes a collapse of the cores during coating (Keller et al., 2022).

### 3.3 Biomass burning emissions

Figure 6 shows the results of measurements of emissions from two logwood stoves during type approval testing. As opposed

to the propane flame aerosol from the CAST, the composition of the wood burning samples is not is not so easily controllable and therefore difficult to reproduce. It depends on the combustion technology and can also have a great degree of variability for samples from a single appliance (see, e.g., Lamberg et al., 2011). Wood burning emissions are composed of OC, EC and non-carbonaceous inorganic materials like ashes and salts. The propane flame examples described in this manuscript use low mass loads, which are relevant for ambient monitoring. The current example shows that FATCAT can measure mass loads in



the hundreds of micro grams of carbon. The upper measurement range of the $CO_2$ sensor (i.e., nominal limit 20,000 ppm) seems to be the limiting factor for these type of emission measurements. During the analysis of these samples, the highest filter load caused a $CO_2$ concentration that shortly surpassed this value. As will be discussed below, sample homogeneity also plays a role as homogeneous samples result in a narrow fast-thermogram that surpasses the upper range of the $CO_2$ sensor at lower filter loads compared to heterogeneous samples with wider thermograms. The wood burning samples create narrow

thermograms.

    The comparison against the standard TOT analysis shows an excellent correlation for TC ($R^2 = 0.996$) for data from two different appliances and two test conditions (i.e., cold and warm start test). The fast thermograms and the details of the TOT fractions (figures 6b and 6c) show the diversity of the samples. EC is the main component, but it evolves differently during analysis, mostly during the EC2 ($550°C$) step for the first sample (i.e., cooking stove, cold start) and on EC3 ($700°C$) for the

other two (i.e., cooking stove and old stove, warm start). On the other hand, EC from the propane flame samples has higher refractor temperatures and evolved mainly during the final EC4 step ($850°C$). The fast thermograms follow this trend and present further nuances. The cold start sample is the first to evolve, followed by the warm cycle of the cooking stove and finally the warm cycle of the old stove. EC from propane flame samples evolved even later in the analysis. In turn, the organics from samples C3 and the coating experiments (M1 coatings 1 through 3) evolve at lower temperatures than the wood burning

samples. Similar to the propane flame samples, the fast-thermograms of the wood burning samples have different degrees of homogeneity which in turn suggests further differences in composition.

    These examples show that fast thermograms contain more information than a simple quantification of TC. The temperature profile is related to volatility and refractoriness of the sample components. Furthermore, they are reproducible and filter-load independent. A coating process, which also affect the structure of the seed particle, adds a new component to the thermogram

without affecting the signal from the uncoated seed. Nevertheless, there are no discrete steps like the ones defined by thermal optical protocols. This poses a new challenge for the interpretation and comparison of the data, specially since thermograms components from heterogeneous samples are not well separated. This is to be expected since carbonaceous aerosol is collection of very diverse substances with a continuum of physical and chemical properties. Conversely, we are not defining a discrete separation in large arbitrary groups which, like the ones from the thermal-optical methods, fail to provide a clean separation

of molecular components (Diab et al., 2015). It is still not clear what information can be extracted from fast thermograms. But they seem to present a reliable and cost-effective opportunity to gather more information from real world carbonaceous samples.

## 4   Conclusions

We developed a novel method for the quantification and characterization of carbonaceous aerosol based on the measurement of

total carbon. Our prototype uses a rigid metallic filter to capture an aerosol sample which is then analyzed by heating the filter through induction to a temperature around $800°C$ in less than a minute under an oxidizing atmosphere. This is long enough to desorbe and/or oxidize all the carbonaceous material of the sample. Full oxidation to carbon dioxide is achieved downstream



of the filter by means of an oxidation catalyst. Quantification is performed by means of a carbon dioxide NDIR sensor. The components selected for this prototype address several downsides of other measurement systems for TC. In particular, the

metallic filter allows for continuous measurement without filter replacement for long periods of time and avoids artifacts caused by, e.g., leakage, displacement, or damage of the sampling filter. The catalyst, in turn, prevents underestimation of the carbonaceous content of the organic fraction due to incomplete oxidation. This combination of components is, to the best of our knowledge, unique to our system.

The limit of detection of the prototype in terms of filter load is $\mathrm{LoD} = 0.19\ \mu\text{g-C}$. In terms of ambient concentrations, this

translates to a $\mathrm{LoD} = 0.32\ \mu\text{g-C/m}^3$ or $\mathrm{LoD} = 0.16\ \mu\text{g-C/m}^3$ for a one and two hours of sampling at 10 lpm respectively. Thus, the method is also suitable for continuous TC measurement in our environment. In a future publication (Keller et al., in preparation), we will report on several months of use at various locations (urban to high alpine) in Switzerland. The upper limit of the measurement technique is set by the upper range of the $CO_2$ sensor and is higher for heterogeneous samples than for homogeneous samples. We have currently successfully measured filter loads up to $\mathrm{TC} \approx 550\ \mu\text{g-C}$. There is an excellent

correlation between our system and other well-established measurement systems. In particular, we compared the TC measured by FATCAT against total aerosol mass measured by means of a TEOM for stable laboratory samples consisting exclusively of carbonaceous material. We also compared FATCAT against TC from TOT analysis for biomass burning emissions from two batch operated logwood stoves.

Another unique feature of our system is the generation of fast thermograms that contain information about the volatility

and refractoriness of carbonaceous particles. Components like SOM, primary OC, and soot evolve at different times during the analysis. This is analogue to thermograms generated through thermal optical methods but without imposing an artificial separation of the carbonaceous material into arbitrary subfractions and without the need for different analysis gases. The use of an oxidizing atmosphere during the heating process prevents the production of pyrolytic carbon, which is a main source of uncertainty in thermal-optical methods. Samples from wood burning emissions or propane flame soot that would appear in the

same subfraction of a thermal optical analysis can be distinguished through additional nuances in the fast thermograms. This feature will be studied further with a long-term employment of FATCAT for ambient air monitoring, where fast-thermograms contain information about the aerosol composition which could be used for source apportionment studies.

*Author contributions.*  AK, PSpecht and PSteigmeier designed and build the prototype. PSteigmeier developed the embedded software and AK developed the controlling and data analysis software. AK designed the experiments and carried them out. AK prepared the manuscript

with contributions from all co-authors.

*Competing interests.*  The authors declare that they have no conflict of interest.

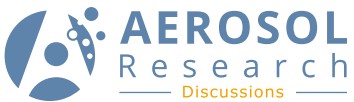

*Acknowledgements.* This work was financed by the Swiss federal office for the environment and the Swiss federal office for meteorology through the GAW-CH Plus research projects 2018-2021. The coating experiments were performed under the umbrella of the 18HLT02 AeroTox and 16ENV02 Black Carbon projects from the European Metrology Programme for Innovation and Research. The authors want

to thank Konstantina Vasilatou (METAS), Daniel Kalbermatter (previously at METAS), Erich Wildhaber (previously at FHNW) and Josef Wüest (FHNW) for their support during the measurement campaigns.





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

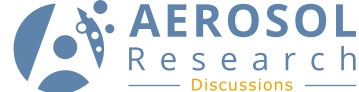



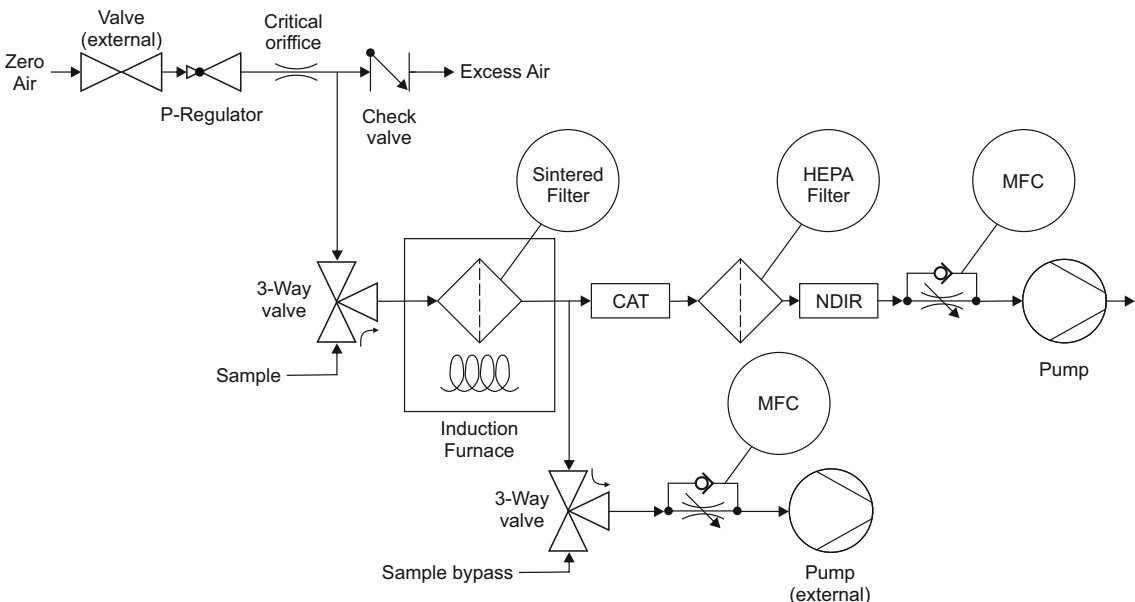

**Figure 1.** Flow diagram of FATCAT. The instrument has three inlets (zero air, sample, and bypass) and three outlets (excess zero air, internal pump and external pump). The three way valves are actuated together so that either the sample or the bypass inlet is open (sampling mode shown here). The flow of zero air is regulated by means of a pressure reducing valve (P-Regulator) and a critical orifice. CAT stands for platinum catalytic converter, NDIR is a $CO_2$ nondispersive infrared sensor, and MFC stands for mass flow controller. Both external components (i.e., the zero air valve and the external pump) are actuated by FATCAT.





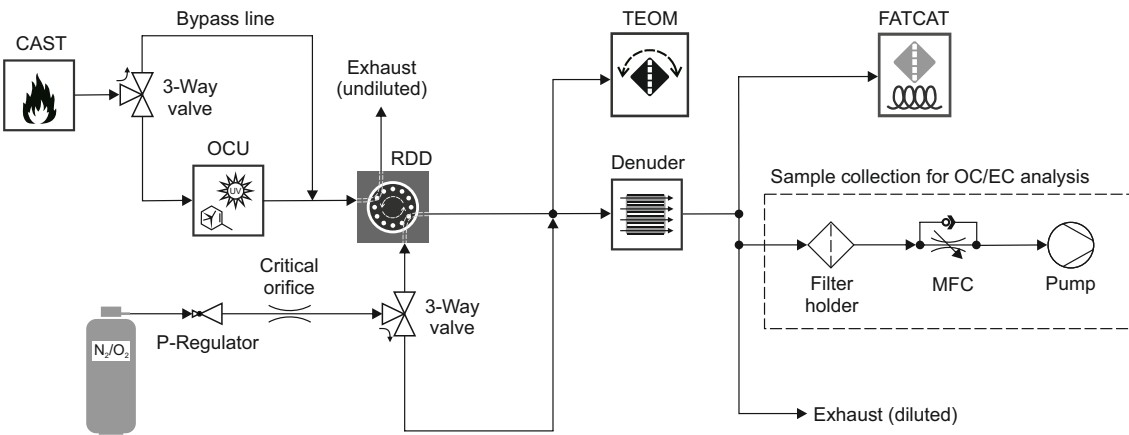

**Figure 2.** Schematic representation of the setup used for the production and measurement of carbonaceous aerosol. The three way valves control the coating of the particles (uncoated mode shown) and use delivery of synthetic air (shown in the diagram) or diluted test aerosol to the measurement devices. The boxes represent the following instruments: combustion aerosol standard soot generator (CAST), organic coating unit (OCU), rotating disc diluter (RDD), tapered element oscillating microbalance (TEOM), and FATCAT. The flowrate of synthetic air ($N_2/O_2$) is regulated by means of a pressure reducing valve (P-Regulator) and a critical orifice. Denuder stands for activated carbon denuder and MFC for mass flow controller.



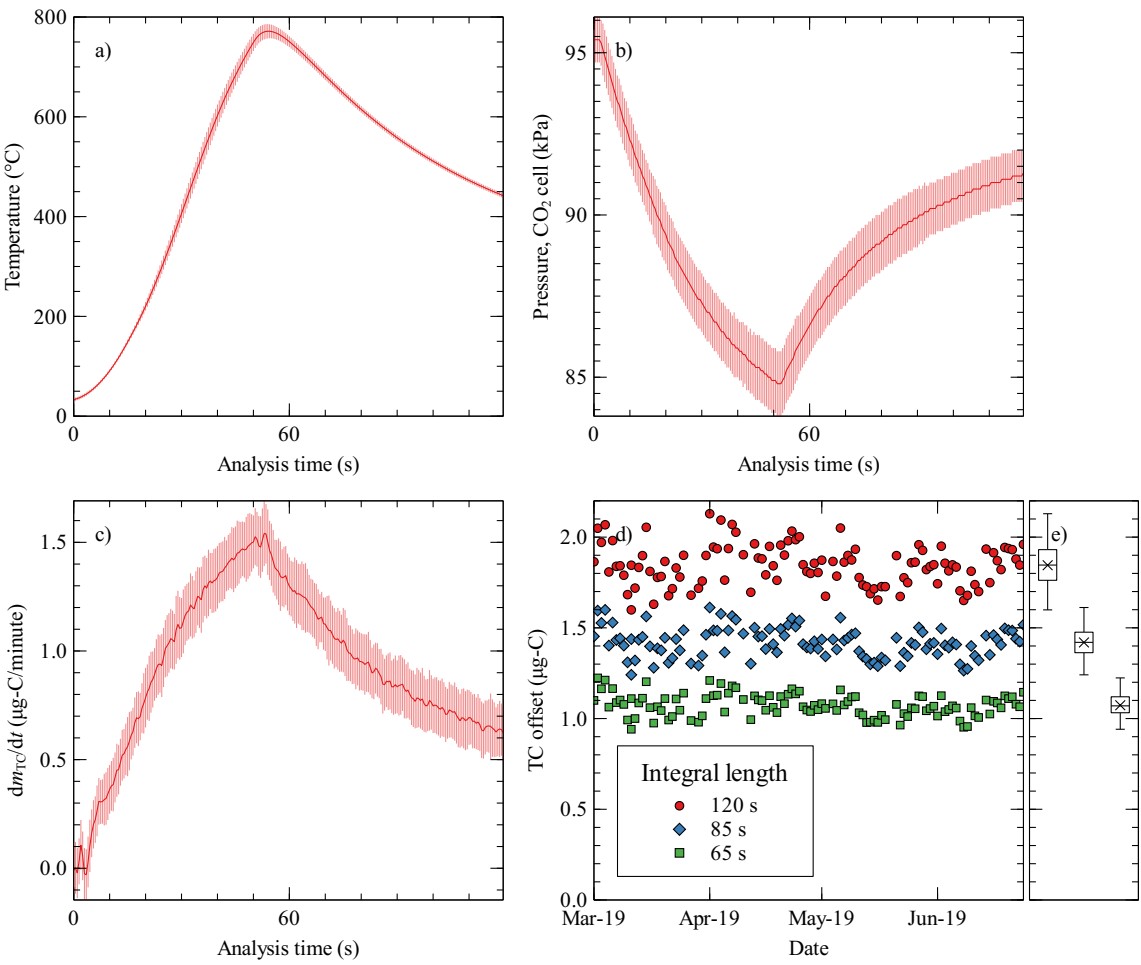

**Figure 3.** Evaluation of the instrument response during the analysis of 109 consecutive blank samples measured once a day during almost four months. Average curve (dark red line) with error bars (standard deviation, light red) for the evolution during the analysis cycles of a) the temperature measured downstream of the sampling filter, b) pressure in the $CO_2$ sensor cell, and c) differential total carbon offset. Time zero marks the start of the heating of the filter. d) The time evolution of the determined offsets in total carbon, i.e. the area under the curve of individual blank measurements that build figure c), and the boxplot representation of the whole series (e) calculated from the integral of the individual blank samples for different integration lengths , starting at time zero of the analysis cycle. μg-C stands for micrograms of carbon.



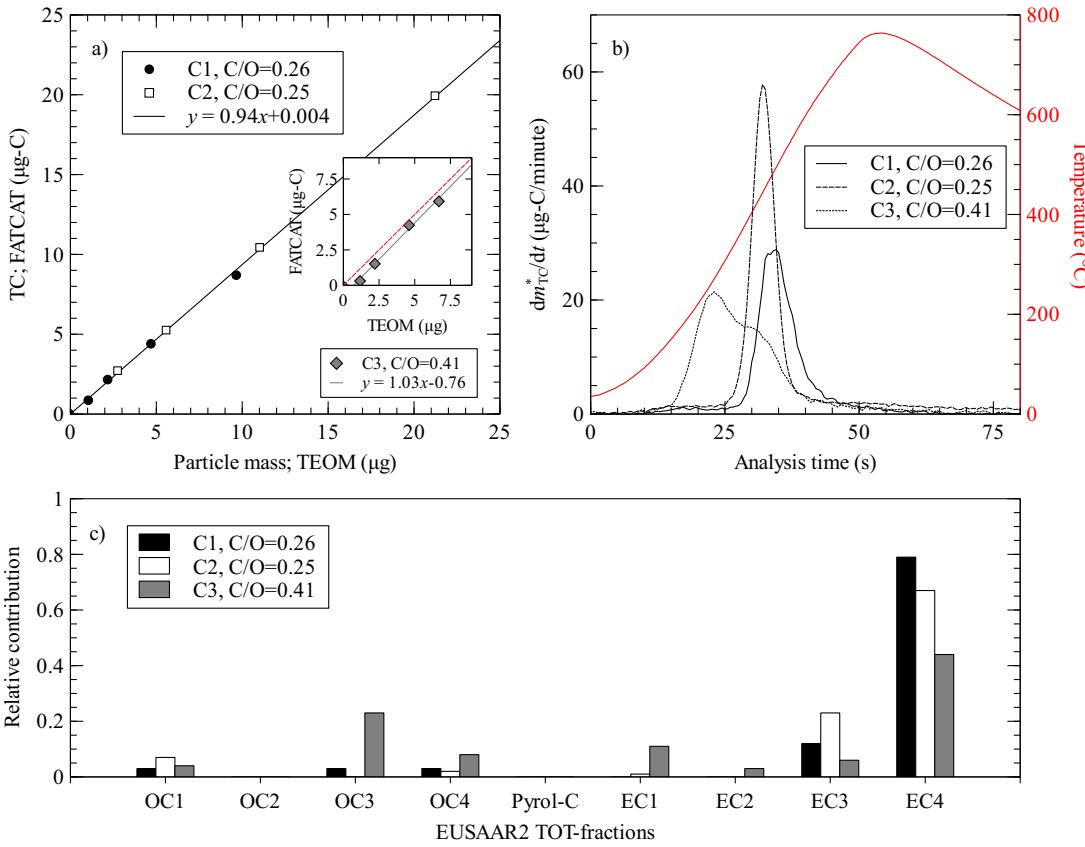

**Figure 4.** a) Total carbon measured by FATCAT against particle mass deposited in the TEOM for 3 different propane flame aerosol samples. The line shows a linear fit to the C1 and C2 samples (coefficient of determination, $R^2 = 0.999$). The insert shows the third sample, C3, a linear fit ($R^2 = 0.993$), and the 1:1 line (dashed lines). b) Blank-corrected fast-thermograms for the 3 types of propane flame samples. Samples with a similar total carbon mass were selected for this comparison, i.e., TC=5.2 µg-C, TC=4.4 µg-C and TC=5.6 µg-C for C1, C2, and C3 respectively. The red line shows the temperature measured behind the filter during the analysis process. c) Relative carbon mass contributions to the analysis steps of the thermal protocol EUSAAR2 for the three samples. OC1 through OC4 are the organic carbon fractions, Pyrol-C is the pyrolysed organic carbon, and EC1 through EC4 are the elemental carbon fractions. C/O refers to the air-to-fuel mixture used to produce the sample.



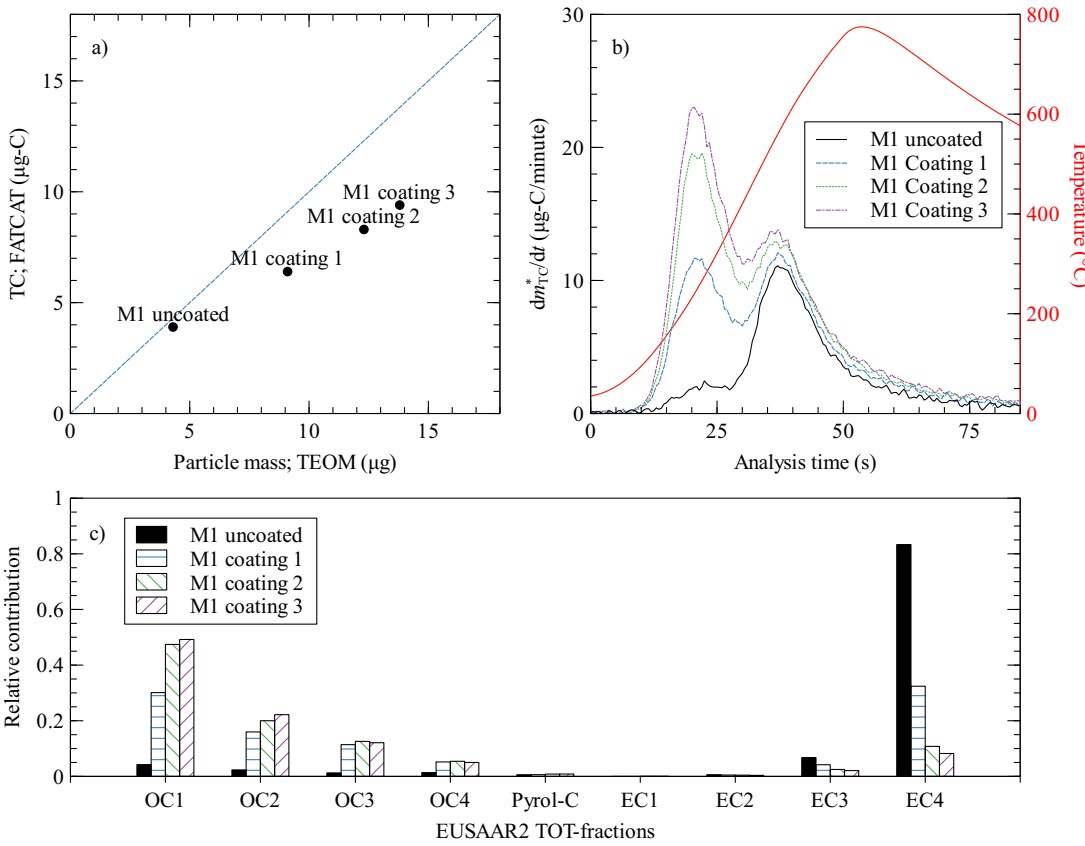

**Figure 5.** a) Total carbon measured by FATCAT against particle mass deposited in the TEOM for the uncoated soot sample M1 and the three different coated samples. The concentration of the seed aerosol was kept constant for all samples. Thus, the increase in mass comes from organic coating. This causes the datapoints to deviate increasingly from the 1:1 line (dashed line), as only a fraction of the mass of organics comes from the carbon atoms. b) Blank-corrected fast-thermograms for the uncoated soot and the three coating levels. The red line shows the temperature measured behind the filter during the analysis process. c) Relative carbon mass contributions to the TOT analysis steps of the EUSAAR2 protocol for the uncoated aerosol and the coated samples. OC1 through OC4 are the organic carbon fractions, Pyrol-C is the pyrolysed organic carbon, and EC1 through EC4 are the elemental carbon fractions.



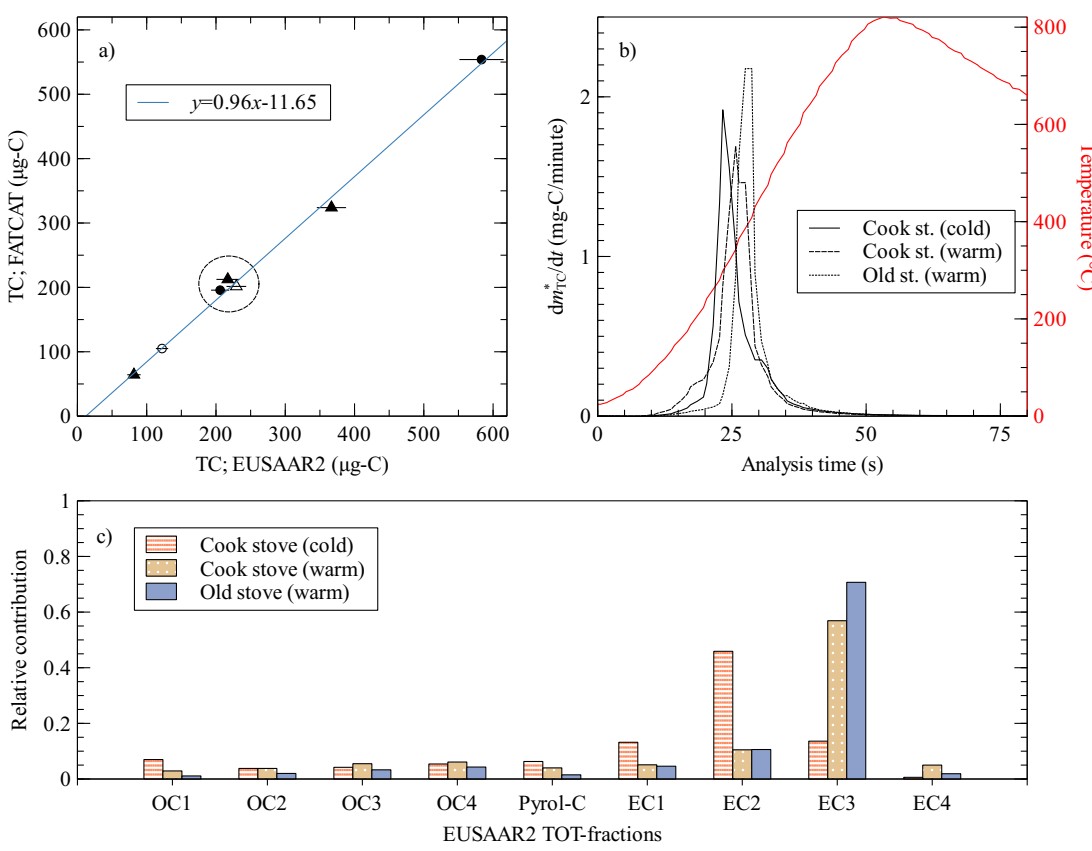

**Figure 6.** a) Total carbon measured by FATCAT against the equivalent, filter size and flow corrected, total carbon from the TOT analysis for wood burning emissions. Triangles and circles correspond of samples from the cooking stove and the old stove respectively. Open and filled symbols correspond to the cold start and warm start tests respectively. The line shows a linear fit to the experimental data. The three points inside the circular region were selected for comparison in the two following graphs of the figure. The error bars of the TOT analysis are based on the uncertainties given by the instrument. b) Fast-thermograms, not blank corrected, for three samples with similar TC. "Warm" means that the data corresponds to a warm cycle of the stove (i.e., a refueling experiment), whereas "cold" correspond to the first heating cycle of the day. The curves are less smooth compared to other figures because the data comes from an earlier stage of the prototype with coarser data resolution. The red line shows the temperature measured behind the filter during the analysis process. c) Relative carbon mass contributions to the TOT analysis steps of the EUSAAR2 protocol for three selected samples. OC1 through OC4 are the organic carbon fractions, Pyrol-C is the pyrolysed organic carbon, and EC1 through EC4 are the elemental carbon fractions.





**Table 1.** Set points for the laboratory generated carbonaceous sample. CAST stands for the combustion aerosol standard generator (Jing Ltd, Switzerland) model CAST-00-4, whereas miniCAST stands for the model miniCAST 5201. The set M1 was used during the coating experiments described by (Kalbermatter et al., 2022). C/O ratio refers to the air-to-fuel mixture during combustion and not to the elemental composition of the produced aerosol.

| Set | Generator | Fuel Propane (mlpm) | Air Oxidation (lpm) | N2 Mixing (mlpm) | Air Mixing (mlpm) | N2 Quenching (lpm) | Air Dilution (lpm) | Overall C/O ratio (-) |
|-----|-----------|------|------|------|------|------|------|------|
| C1 | CAST | 56.7 | 1.534 | 240 | – | 7.62 | 18.0 | 0.26 |
| C2 | CAST | 47.0 | 1.320 | – | – | 6.47 | 16.5 | 0.25 |
| C3 | CAST | 47.0 | 0.685 | – | – | 6.47 | 16.5 | 0.41 |
| M1 | miniCAST | 60.0 | 1.100 | – | 220 | 7.00 | 10.0 | 0.28 |



**Table 2.** Specifications of the two batch-operated logwood appliances used for this study. n.a. stands for not available. kw stands for kilowatts.

| ID | Type | Manufacturer | Model | Nominal power | Year |
|----|------|--------------|-------|---------------|------|
| Cook Stove | Cooking Stove & furnace | TL-Tech, Switzerland | Reiat | 8.5 kw | 2016 |
| Old stove | Chimney stove | Jøtul, Norway | F 3 | 6.8 kw | n.a. |



**Table 3.** Offset of the average, uncorrected, blank sample from the data shown in figure 3 for 3 different integration lengths. The parentheses shows the standard deviation. LoD is the corresponding limit of detection calculated from the noise-to-background using the $3\sigma$ criterion. µg-C stands for micrograms of carbon.

| Integral length (s) | TC offset (µg-C) | LoD (µg-C) |
|---|---|---|
| 120 | 1.85 (0.11) | 0.34 |
| 85 | 1.42 (0.08) | 0.25 |
| 65 | 1.07 (0.06) | 0.19 |

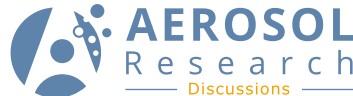

**Table 4.** Properties of the propane flame aerosol samples. The M1 samples correspond to the aerosol generated and described by Kalbermatter et al. (2022) as set-point 0.1. The uncertainty of the EC/TC is based on the uncertainties given by the instrument's software, calculated as the detection limit of $0.2\,\mu\text{g-C}\cdot\text{cm}^{-2}$ plus 5% of the carbon mass determined in the analysis for each carbon fraction.

| Set | Coating | EC/TC mass fraction (%) | GMD (nm) |
|-----|---------|-------------------------|----------|
| C1 | uncoated | $91 \pm 17$ | 72 |
| C2 | uncoated | $91 \pm 19$ | 150 |
| C3 | uncoated | $65 \pm 11$ | 29 |
| M1 | uncoated | $84 \pm 8$ | 88 |
| M1 | coating 1 | $37 \pm 4$ | 90 |
| M1 | coating 2 | $13 \pm 1$ | 111 |
| M1 | coating 3 | $10 \pm 1$ | 126 |