# Peer review of "A novel measurement system for unattended, in-situ characterisation of carbonaceous aerosols"

_Aerosol Research, 2023_

## Author Comment (AC1)

The authors would like to thank the reviewers for their valuable input. Their perspectives and comments have helped to improve the content of the manuscript. The responses below address all the points raised by the reviewers. Our responses are shown in blue, deleted text in red, and added text in gold.

**Points raised by RC1:**

Suggestions to clarify the discussion.

1. I did not understand why the authors claim that FATCAT must be operated with an analytical synthetic air gas. The CO2 background of ambient air cannot possibly change fast enough to cause baseline problems. Why not just fit and subtract a baseline? The Magee TCA08, a similar instrument, does this.

    We have added the following text to the description of the instrument:

    > "It could be possible to use ambient air for the analysis and subtract the CO2 baseline as described below. Nevertheless, further characterization is needed to determine how this would affect parameters like the limit of detection of the instrument."

2. The useful range of the instrument is "above baseline CO2" and "below saturation of the CO2 detector". This means that any lower detection limit corresponds also to an upper loading limit. It may be clearer to state this in the abstract? Then the reader can understand the "operating range" available at a given flow rate and sampling duration. Also, Section 3.3 discusses the upper limit in a confusing way, mentioning only the CO2 sensor's range and not the loading parameters (flow and sampling duration). This could be clarified.

    While the measurement range of FATCAT is closely related to the range of the CO2 sensor, other parameters, beyond the CO2 sensor range and the sample flow rate, come into play. We believe that adding the CO2 discussion to the abstract will not serve the understanding of the limitations. Instead, we have moved the discussion of the upper and lower limits of the instrument to section 2.1. The text reads:

    > "By taking a closer look at equation 2, it becomes clear that the measurement span of FATCAT is closely related to the performance of the $CO_2$ sensor. The limit of detection of the sensor and the length of the integral, together with the already mentioned variations of the $CO_2$ baseline, will directly affect the limit of detection of FATCAT. On the other end, FATCAT's upper limit of quantification is determined by the upper measurement range of the $CO_2$ sensor (i.e., nominal limit 20,000 ppm) as well as by the shape of $c_{CO2}(t)$. We will show that $c_{CO2}(t)$ produces curves that can be interpreted as thermograms. As will be discussed below, homogeneity of carbonaceous species plays a role as homogeneous samples result in narrow thermograms that may surpass the upper range of the $CO_2$ sensor at lower filter loads compared to heterogeneous samples with wider thermograms."

> "The current example demonstrates that FATCAT is capable of measuring mass loads in the hundreds of micrograms of carbon. The analysis of the highest filter load of TC=554 ug-C caused a brief $CO_2$ concentration exceeding the $CO_2$ sensor's quantification limit. This may be considered a worst-case upper detection limit as the wood burning samples in this study produced narrow fast thermograms."

3. (3a in RC1) Line 43: "per definition, only organic aerosol contains carbon-hydrogen bonds" but a few lines later "even soot...contains hydrogen and other elements". The first statement should be removed, there is no definition of organic aerosol like this. The second statement should be updated to "even soot ... contains oxygen and hydrogen" because the oxygen is a much more substantial mass fraction. Then at line 259 the authors state, "by definition, elemental carbon consists exclusively of carbon atoms" which is in contradiction to their earlier acknowledgement that elemental carbon always contains hydrogen (and oxygen).

It seems like this is a minor error that crept in during revisions of the manuscript. Please remove all statements implying that soot or EC only contains carbon, and ideally add further citations showing that soot contains oxygen and hydrogen.

We have modified the discussion around line 43. The text now reads:

> "One of the most commonly used approaches for classifying carbonaceous aerosol is through thermal refractivity analysis which separates it into the complementary fractions of organic carbon (OC) and elemental carbon (EC). The term "organic carbon" can be misleading, as, in a more general sense, organic compounds are those that contain carbon-hydrogen bonds. Although EC has a high carbon content by weight, even reference EC samples used for calibration purposes contain hydrogen and other elements (Clague et al., 1999). Thus, the main disadvantage of thermal-optical analysis lies in the facts that EC and OC are defined operationally from a sample's behavior during analysis and do not refer to a well-defined material (Corbin et al., 2020, and references therein). Other common carbonaceous fractions include equivalent black carbon (eBC), which is measured by light absorption, and refractive black carbon (rBC), which is measured by laser-induced incandescence (Petzold, et al., 2013; Stephens et al., 2003). Some aerosols like soot can be classified as EC, eBC, and rBC. However, these definitions are not interchangeable, as each fraction cannot be inferred definitively from one another."

Other appropriate modifications are shown in the answer to RC1 3b.

4. (3b in RC1) In fact, the FATCAT measures TC and the TEOM measures PM. Therefore, the comparison of FATCAT and TEOM has provided a measurement of the mass fraction of carbon in the sample, as mentioned in #3a above. The authors' slope of

Figure 4a, 0.94, means that 94% of the sampled mass was carbon and 6% was oxygen plus hydrogen. This 94% compares well with the values of 90% and 93% reported by Corbin et al. (2020), and the values of 90% to 98% quoted in that paper for other literature studies. Therefore, FATCAT/TEOM appears to be an accurate technique for measuring this important quantity, which is required for converting TC measurements to PM mass!

The authors mention that the 94% carbon fraction is close to the EC/TC of 0.91. That is true, but this is purely a coincidence. Both EC and OC = TC - EC contain carbon and would be measured by FATCAT. This is also explained in Corbin et al. (2020) Equation 6.

By the way, I recommend changing from "91%" to "0.91" for the EC/TC here and in the tables, because it may confuse readers. The sample contained 94% carbon, and that carbon was divided into operational parameters EC and OC. The EC/TC definition is not a physical one and not even thermodenuded soot is "100% EC".

The mass fractions are now consequently expressed without the percent sign throughout the manuscript.

We also changed the relevant text from:

> "By definition, elemental carbon consists exclusively of carbon atoms and, thus, this fraction should be measured equally by the two instruments. The slope of the correlation, m=0.94, is very close to the EC/TC=0.91 calculated using TOT. The difference could be explained by the carbon mass provided by the OC fraction of the samples."

To:

> "The slope of the correlation, m, indicates that the sample has a carbon mass fraction of $f_C$ = m = 0.94. This compares well with the values of $f_C$ = 0.90 and $f_C$ = 0.93 reported for other flame generators (Corbin et al., 2020), and the values of $0.90 \leq f_C \leq 0.98$ quoted in that manuscript for other literature studies."

We highlight the fc=TC/TEOM method in the conclusions. We modified this text:

> "There is an excellent correlation between our system and other well-established measurement systems. In particular, we compared the TC measured by FATCAT against total aerosol mass measured by means of a TEOM for stable laboratory samples consisting exclusively of carbonaceous material. We also compared FATCAT against TC from TOT analysis for biomass burning emissions from two batch operated logwood stoves."

Replacing it by:

> "FATCAT was validated against aerosol mass measurements using TEOM and against TC from TOT analysis. Experiments carried out with standard laboratory-generated samples and batch operated logwood stoves' emissions displayed a high level of correlation between these methods. We demonstrate that the combination of TC calculated using FATCAT with measurements of aerosol mass can serve as a technique for evaluating the carbonaceous fraction, $f_C$, of aerosol samples."

5. (was RC1 4) It is only the Magee TCA08 that would define OC as complementary to eBC. The eBC definition of Petzold should be quoted here, which is a consensus definition. The danger of defining TC = eBC + OC is clear: coatings can cause eBC to be up to 2x larger due to "lensing" or absorption enhancement, and then the definition breaks down. I recommend quoting Petzold et al. (2013) and avoiding partial definitions to avoid confusion.

We removed the text suggesting that TC can be defined as TC = eBC + OC (line 43, see answer to RC1, 3a). The concept equivalent organic carbon (eOC) is still mentioned in the introduction when referring to the TCA08.

6. (was RC1 5) I was surprised that the authors did not emphasize the improved sensitivity of their device in the introduction. The device has a much lower limit of detection than the thermal-optical analysis to which the authors compare it. This is a significant benefit for e.g. instrument calibration. The FATCAT is so sensitive that it probably cannot sample in parallel to thermal-optical filter samplers.

The TOA Model 4 Semi-Continuous OCEC Field Analyzer from Sunset, which uses the complete sample filter during analysis, advertises a limit of detection (LoD) that is similar to FATCAT's LoD. Nevertheless, we believe that our method offers other advantages that are already discussed in the manuscript.

7. (was RC1 6) I believe that it is not the use of an oxidizing atmosphere that prevents pyrolysis (line 368), but the rapid heating protocol. Pyrolysis reactions occur in competition with evaporation and oxidation, and rapid heating means that pyrolysis "loses" the competition. The authors have stated the opposite. I request that the authors add a citation here (or earlier in the manuscript) supporting their opinion, or remove this statement in the absence of clear evidence.

We removed the statement from the conclusions as well as from section 3.2.

**Minor comments**

Line 68, "Problems like..." were these issues shown to be the cause of problems, or were they simply listed speculatively? I do not believe that the listed issues are demonstrated problems in TOA. Please cite clear evidence if I am wrong, or remove the list of speculative statements.

The cited reference "Panteliadis et al., 2015" discusses variations between instruments and protocols during an intercomparison campaign, including these sources of systematic errors. We have rephrased the text from:

> "Problems like dependency of measurement day, variations in flow-rate within the accepted operation range, variations in the calibration gas (i.e. when changing the gas bottle) or in transit time through the instrument, leakages, and different rates of pyrolyzed OC production were reported as sources of unresolved systematic errors."

To:

> "Dependency of measurement day, variations in flowrate within the accepted operation range, variations in the calibration gas (i.e. when changing the gas bottle) or in transit time through the instrument, leakages, and different rates of pyrolyzed OC production were reported as sources of unresolved systematic errors."

line 51 onwards only defines thermal refractivity methods in the atmospheric sciences. Materials and other (e.g. soil) scientists also use thermal refractivity methods. Please add "In atmospheric science" before this paragraph.

Added.

Line 74 states that "Light absorption methods for measuring eBC are prone to systematic errors". This is not true, light absorption can be measured accurately and without error (note that "lensing" effects are not errors, but physical phenomena). The authors only cite studies on filter-based attenuation photometers after this statement. Perhaps the authors meant to state that filter-based attenuation is prone to systematic errors?

Changed from:

> "Light absorption methods…"

To:

> "Filter-based light attenuation methods…"

Line 99, "all OC" should be "all TC". For example, carbon monoxide is not OC.

Changed

Line 128, give pressure of sampling site, not just m.a.s.l.?

Added. The text now reads:

> "…the Swiss plateau (approximate elevation of about 400 meters above sea level [m.a.s.l.], ambient pressure around 960 mbar), and around 7 lpm at the Sphinx observatory of the Jungfraujoch (situated at 3500 m.a.s.l., ambient pressure around 640 mbar)."

Line 174 a comment about the setup for another experiment does not seem to belong here.

It is relevant because FATCAT was not connected to an aerosol generator. As mentioned in the text above the comment, we wanted to state that the FATCAT was not simply idle between each baseline measurement.

Line 193 C1 C3 M1, are not yet defined and the heading doesn't match the text.

The setpoints are defined in table 1, which in turn is cited in line 184.

Line 211 please expand on "to increase the oxygen content". I understand that FATCAT needs it for combustion, but how does the user know how much dilution is needed?

For this manuscript, and for FATCAT's operation, the oxygen content is not relevant. We have adapted the text from:

"to increase the oxygen content and avoid condensation of water"

To:

"to avoid condensation of water"

On line 268 the authors speculate that the TEOM adsorbed OC3. My feeling is that this is unlikely. See Subramanian et al. (2004, https://doi.org/10.1080/02786820390229354). The speculation here should be removed or made more quantitative, for example, it currently compares the 50 C TEOM with the 250 C OC3 -- totally different temperatures? In fact the results appear quite accurate (see #3b above) but the importance of the problems discussed by Subramanian should indeed be assessed. Ideally, in a separate section.

We noticed a typo in our manuscript. OC3 evolves (desorbs) at 450 C (not 250 C). This has been corrected in the manuscript. The TEOM filter heated at 50 C may prevent positive artifacts from high volatility organic carbon or from water but not from the low volatility organic species that correspond to OC3.

Subramanian (2004) points out the importance of positive artifacts in non-denuded samples. As a matter of fact, figure 9a of their manuscript compares two 24-hour samples, diluted and undiluted, that have remarkable fit with a slope close to one but a non-zero intercept. The authors conclude that this is caused by a constant positive artifact arising from the fact that the 24-hour filter has reached equilibrium with the gas phase. This corresponds to the data shown in the inset of figure 4a of our manuscript. The slope is close to one and the non-denuded line (i.e., the TEOM) has an apparently constant positive artifact. Filter equilibrium will be reached much faster in our case as we are exposing the filters to concentrations that are several orders of magnitude higher than ambient concentration.

We had changed the relevant fraction of the text from:

> "A positive artifact from the TEOM, due to the retention of gas phase organic species, could cause the displacement of the intercept and the steeper slope."

To:

> "A positive artifact from the TEOM, due to the retention of gas phase organic species from the non-denuded sample, could cause the displacement of the intercept. Similar behavior has been demonstrated through a comparison of denuded and non-denuded samples. (Subramanian et al., 2004)"

Line 275: there is no need to refer to thermal-optical methods for the term thermograms, the term is used in other techniques as well.

Changed from:

> "We refer to them as fast-thermograms because they resemble thermograms produced by thermal-optical methods. Nevertheless, there is neither a heating protocol based on predefined temperature steps nor a split between EC and OC using different gases."

To:

> "We refer to them as fast thermograms. Nevertheless, as opposed to thermograms produced by thermal-optical methods, there is neither a heating protocol based on predefined temperature steps nor a split between EC and OC using different gases."

Line 288: consider citing other authors in addition to Kelesidis et al., e.g. Maricq (2014, doi:10.1080/02786826.2014.904961).

Suggested source added.

Line 300 and related: I would recommend calculating the OM/OC ratio, a common metric, to place this discussion in better context.

We now use OM/OC in the discussion. Some text modifications are shown in the answer to RC2. Additionally, we replaced:

> "Other oxidation states, leading to diverse contributions of carbon to the total aerosol mass, ..."

With:

> "Other oxidation states, leading to diverse OM/OC..."

Line 306: "This is not evident"? Meaning "the collapse of the soot core did not affect the thermogram"?

Yes. To clarify this, we substituted:

> "This is not evident as the organic material causes a collapse of the cores during coating (Keller et al., 2022)."

With:

> "This is not obvious as the organic material causes the cores to collapse during coating (Keller et al., 2022), which in turn could have affected the thermogram feature corresponding to the particle core."

Figure 6 is discussed by reference to Figure 5's thermograms. Therefore, it would be helpful to overlay one Figure 5 thermogram over Figure 6.

All the samples (figures 4 and 5) are used as a reference for the discussion. We could not find a good way of combining all thermograms in figure 6. So we have modified the discussion to cite explicitly the times of where the maximum of the thermograms appear.

The following text was added:

> "The cold start sample is the first to evolve (max dm*$_{TC}$/dt at t = 23 s), followed by the warm cycle of the cooking stove (max dm*$_{TC}$/dt at t = 26 s) and finally the warm cycle of the old stove (max dm*$_{TC}$/dt at t = 28 s). EC from propane flame samples evolved even later in the analysis (max dm*$_{TC}$/dt at t = 34, 32 and 37 s for C1, C2, and M1 respectively). In turn, the organics from the coating experiments evolve at lower temperatures (i.e., organics peak at t = 20.5 s for M1 coatings 1 through 3) than the wood burning samples. C3 shows a special situation, with two peaks that are close together. The first one, with a maximum at t = 23 s, may be a combination of the OC3, OC4 and EC1 components. The time corresponding to the maximum of the second C3 peak cannot be extracted without further analysis, but it is located at t ≈ 30 s and corresponds most likely to the EC4 component."

Line 337: Thermograms of homogeneous samples are also not well separated. Here, reviews of thermo-optical analysis could be cited.

We refer to the (fast) thermograms of this study. We adapted the text from:

"…since thermograms components from heterogeneous samples are not well separated."

To:

"…because fast thermogram features from different samples can be partially overlapping (see, e.g., figure 4b)"

Line 370: I would recommend that the authors consider building a library of source signatures, in order to assist with the future interpretation of FATCAT thermograms for source apportionment.

Indeed, this is an important point.

**Points raised by RC2:**

**General Comment**

Generally speaking, the part dealing with the thermograms of coated/uncoated CAST soot and biomass smoke is most interesting, and the authors are a bit too modest in their claim that "it is still not clear what information can be extracted from the fast thermograms. But they seem to present a reliable and cost-effective opportunity to gather more information from real world carbonaceous samples" (p 11, lines 340 – 342) – just one look at the thermograms shows that carbon from different samples evolves at different times (i.e. temperatures) and gives thermograms of different shapes, which can be used to derive signatures of e.g. Diesel soot (not tested here, but could easily be done), aged particles and biomass smoke. The current use of thermo-optical analyzers does not utilize the info contained in the highly complex thermograms obtained during sample analysis; only numbers on the different OC and EC fractions are given. The new instrument gives quick thermograms that can be analysed fairly quickly (or even automatically – AI could help ….)

Thank you from this remark. We now emphasize the importance of the thermogram further. We changed the relevant text from:

> "It is still not clear what information can be extracted from fast thermograms. But they seem to present a reliable and cost-effective opportunity to gather more information from real world carbonaceous samples."

To:

> "It has not yet been conclusively investigated how much information about the composition can be obtained from the fast thermograms. What is certain is that they offer a reliable and cost-effective way of obtaining more knowledge about real samples containing carbon."

**Small comments**

A few pieces of info are missing.

What is the size of the exposed filter surface area? The LOD's are given in µg C, translated to µg-C*m-³, but it would be interesting to see how this compares to the LOD of thermo-optical methods (usually given in µg-C*cm-²).

This comparison is not relevant for our system as the detection limit in µg C is given by the time needed for heating and the noise of the $CO_2$ sensor, and not by the filter surface. As discussed in the manuscript, the filter surface has an influence on the maximum flowrate

that can be used during sampling. Nevertheless, modifying this is not as easy as, e.g., increasing the filter surface, as this would require an optimization of the induction furnace.

Which types of firewood were used in the test of biomass smoke? Could have an influence on the thermograms. The carbon emissions are different for different fuels (see Sun et al. 2021, Atmos. Chem. Phys., 21, 2329–2341 or Priestley et al. 2023, Environ. Sci.: Atmos., 2023, 3, 717)

We added the sentence:

"Beech wood was used throughout all experiments."

How long does the cleaning of the system with zero air take?

This question is not clear to us. The filter needs no separate cleaning because every analysis cycle fully removes the carbonaceous material.

Cleaning of the filter is only needed after the installation as manufacturing and/or handling may have contaminated the filter. Nevertheless, we do not have enough experience with this, as the current filter showed no sign of degradation after >20,000 hours of operation measuring ambient samples. We added a word about this in the conclusions:

"We still have not determined the typical operation time before filter replacement is needed. Our current configuration has been tested for more than two years of continuous operation, sampling ambient aerosol at different locations in the Swiss plateau, without showing signs of degradation. This may be different for other locations. In any case, filter replacement can be programmed as a standard procedure when replacing other components like the lamp of the NDIR sensor."

Which quartz fibre filters were used for the TOT analyses?

We now include the two types of filters (i.e., QR-100 quartzfibre filters and Pallflex Tissuquartz 2500QAT-UP) and indicate in which campaigns they were used.

Was the same type of zero air used in all experiments?

Yes, we have changed the text in line 130 of the original manuscript from:

"As a standard, we use synthetic air …"

To:

"All experiments described in this manuscript use synthetic air…"

Just a comment: it is really good to see that the effect of filter temperature on pore size is taken into account

[Figure]

A general comment: The English is very good, but please check the whole MS again for correct use of singular/plural "s" and "this" vs. "these"…..

Revised

**Further comments in the order of appearance in the text**

Line 34: the words "sufficient accuracy" can refer only to TC – the whole question about accuracy of determination of EC (or any other component of carbonaceous aerosols) is still open – unless one considers method-specific definitions as "accurate".

The statements made refer to TC, as this parameter is well defined and does not depend on the choice of protocols and methods (as is the case with EC, for example).

We specify this in the paper by adding the reference in to TC in the relevant section:

> "However, there is still no commercial instrument that can measure the totality of carbonaceous aerosol (i.e. aerosol-bound total carbon, TC) with sufficient accuracy and temporal resolution on a global level over extended periods of time in an unattended manner with minimal user intervention."

Line 55: the different protocols vary not only in the number of temperature steps, but also in the temperatures at these steps

Changed to:

> "Different protocols vary in terms of the number of temperature set-points and target temperature that define the subfractions"

Line 58: The EUSAAR2 protocol takes 17 minutes for running through a heating cycle, but the necessary cool-off phase takes another 5-10 minutes. I suspect that the time you give for analyzing samples with the IMPROVE protocol also does not include cooling off times, while sample analysis times with your method include cooling off.

Thank you, this is a good parameter to consider, especially for those that use the field version of the TOA systems.

We added the following sentence:

> "The duration of the protocols does not take into account the cool-down time needed before the device is ready for the next cycle."

Line 92: The statement "... show that these thermograms contain information only about the composition of the aerosol" is too comprehensive – the thermograms contain info only about the composition of the _carbonaceous_ aerosol

We now specify the type of aerosol:

> "...show that these thermograms contain information about the composition of the carbonaceous aerosol"

Line 108: explain acronym FHNW (the authors' address gives the English name of their institution)

Changed to:

> "university of applied sciences and arts northwestern Switzerland (FHNW in German)"

Section 2.3: obviously (see Table 1) both a CAST and a miniCAST were used, but only the CAST is mentioned in the text?

We adapted line 180 of the original manuscript from:

> "The second set of experiments consisted of CAST particles with a high EC-to-TC ratio,…"

To

> "The second set of experiments consisted of particles with a high EC-to-TC ratio, generated by means of a miniCAST 5201 Type BC (Jing Ltd., Switzerland),"

Furthermore, line 184 was changed:

> "The CAST and miniCAST set-points used for both campaigns…"

Additionally, figure 2 now shows the possibility of using a CAST or a miniCAST.

Line 190: a nice typo: "tampered oscillating micro balance"

Corrected:

> "tapered element oscillating microbalance"

Line 243: explain acronym OEM

Added: "original equipment manufacturer (OEM)"

Line 268: change "... gas phase EC form the C3" to "…. from …"

Changed

Line 300: " ... most of the organic material corresponds to OC1...." – this is not supported by the histograms shown in Figure 5c

Rephrased from:

> "In our case, most of the organic material corresponds to OC1, the most volatile fraction of the thermal-optical analysis (figure 5c). This step is performed at 200 C under an helium atmosphere."

To:

> "In our case, OC1 is the predominant organic carbon fraction (figure 5c). This step corresponds to the most volatile fraction of the thermal-optical analysis, performed at 200 C under a helium atmosphere."

Figure 5: contrary to figure 4, no regression line is given. Drawing a line through the three points for the coated sample seems to give a nice zero intercept – only the point for the uncoated sample does not lie on this line. Any ideas why? The slope of the line through the points for the coated samples gives an indication of the mass fraction of carbon in the coated particles. Of course one could argue that four data points are too few to draw conclusions, but conclusions are drawn in the discussion of Figure 4, which also has only four points for each of the C1 – C3

Thank you for this suggestion. We have added a regression line to the 4 points (coefficient of determination, $R^2=0.996$) and calculated the carbon mass fraction of the coating. For this discussion, we also calculated a 3-point fit by leaving out the uncoated data point. The resulting slope and intercept differ only slightly from the 4-point fit (slope ~10% steeper, intercept ~50% smaller), but the intercept is still positive and non-zero.

We added the following text to Fig. 5 caption: "The solid line shows a linear fit to the 4 datapoints (coefficient of determination, $R^2=0.996$)."

And modified the discussion by replacing the text:

"Carbon is only a fraction of the coating material, as SOM produced by ozonolysis also contains hydrogen and oxygen. Thus, the mass fraction of carbon to the total mass of the particles is reduced by every further coating step (figure 5a). As an example, by looking at the oxidation state of carbon, we can determine that carbon may amount to only 55% of the total mass of a-pinene SOM produced using a similar setup (Leni et al., 2022, supplementary information)."

With:

"Here again, the carbonaceous fraction of the coating material can be inferred from the slope of the linear regression between TEOM and FATCAT (figure 5a) to $f_c = 0.56$. The inverse of $f_c$ indicates that the coating material has a ratio of OM/OC = 1.79. This is in excellent agreement with the range of $1.78 \leq$ OM/OC $\leq 1.85$ reported for α-pinene SOM produced using a similar setup (Leni et al., 2022, supplementary information)."

Line 366: change "analogue" to "analogous"

Changed

Author contributions: EW is not mentioned?

We now state EW's contribution explicitly by:

"AK prepared the manuscript with contributions from EW and all other co-authors."

References: Szopa et al refers to a chapter in the new IPCC report – which chapter?

Indeed, this is chapter 6 "Short-Lived Climate Forcers". This is already included in the BibTeX file that will be uploaded. The produced reference follows the citing recommendation of the

IPCC (see
https://www.ipcc.ch/report/ar6/wg1/downloads/report/IPCC_AR6_WGI_Citation.pdf)